# Explainably Safe Reinforcement Learning

**Sabine Rieder**[*] 🆔
Masaryk University
Technical University of Munich
sabine.rieder@mail.muni.cz,

**Stefan Pranger**[*] 🆔
Graz University of Technology
stefan.pranger@tugraz.at

**Debraj Chakraborty** 🆔
Masaryk University
chakraborty@fi.muni.cz

**Jan Křetínský** 🆔
Masaryk University
Technical University of Munich
jan.kretinsky@fi.muni.cz,

**Bettina Könighofer** 🆔
Graz University of Technology
bettina.koenighofer@tugraz.at

## Abstract

Trust in a decision-making system requires both safety guarantees and the ability to interpret and understand its behavior. This is particularly important for learned systems, whose decision-making processes are often highly opaque. Shielding is a prominent model-based technique for enforcing safety in reinforcement learning. However, because shields are automatically synthesized using rigorous formal methods, their decisions are often similarly difficult for humans to interpret. Recently, decision trees became customary to represent controllers and policies. However, since shields are inherently non-deterministic, their decision tree representations become too large to be explainable in practice. To address this challenge, we propose a novel approach for explainable safe RL that enhances trust by providing human-interpretable explanations of the shield's decisions. Our method represents the shielding policy as a *hierarchy of decision trees*, offering top-down, case-based explanations. At design time, we use a world model to analyze the safety risks of executing actions in given states. Based on this risk analysis, we construct both the shield and a high-level decision tree that classifies states into risk categories (safe, critical, dangerous, unsafe), providing an initial explanation of why a given situation may be safety-critical. At runtime, we generate localized decision trees that explain which actions are allowed and why others are deemed unsafe. Altogether, our method facilitates the explainability of the safety aspect in the safe-by-shielding reinforcement learning. Our framework requires no additional information beyond what is already used for shielding, incurs minimal overhead, and can be readily integrated into existing shielded RL pipelines. In our experiments, we compute explanations using decision trees that are several orders of magnitude smaller than the original shield.

## 1 Introduction

*Deep reinforcement learning* (RL) [42] is a powerful machine learning technique for intelligent sequential decision-making. Despite its successes, its application in safety-critical systems remains limited due to safety concerns. RL agents learn by exploring their environment through trial and error, a process that inherently carries the risk of taking unsafe actions.

39th Conference on Neural Information Processing Systems (NeurIPS 2025).

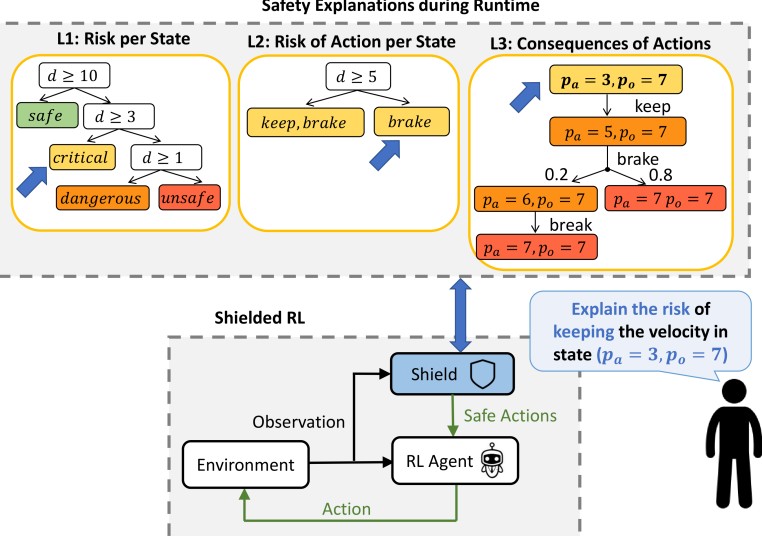

Figure 1: Overview of our method for explainable-safe RL.

*Shielding* [1] is a prominent approach towards safe RL. A shield blocks ("shields") unsafe actions from the learning agent at runtime, either during learning or evaluation, as depicted at the bottom of Fig. 1. At each step, the shield provides a list of all safe actions, from which the agent can select one. Shielded RL thus belongs to a class of methods that combine symbolic AI, which provides formal safety guarantees but suffers from limited scalability, with machine learning, which offers high scalability but lacks guarantees. However, even though shields add safety guarantees to RL, trusting a shielded system remains challenging due to the lack of *explainability*.

Policies learned via RL are notoriously difficult to interpret, as they are represented by highly opaque deep neural networks. Unfortunately, understanding the decisions made by a shield is almost equally challenging. Shields are typically implemented as large lookup tables that define the set of allowed actions for each state. These tables are inherently hard for humans to interpret, as they provide no insight into why a particular action is considered safe or unsafe in a given state. The combined opacity of both the shield and the RL policy makes it difficult to predict, understand, or trust the behavior of a shielded learning agent.

In this paper, we propose an approach to *explainably safe RL* by explaining the decisions of the shield to the user, making the safety aspect of the process more explainable.

*Decision Trees* (DTs) have recently gained popularity for representing controllers and policies due to their inherent simplicity and human-readability [17]. This is documented by numerous case studies and tool support, e.g. [5, 4, 7, 12]. However, directly computing DTs to represent a shield often results in excessively large trees, undermining their utility for explainability. Computing compact DTs is especially challenging in the context of shielding due to the shield's inherent non-determinism: a shield permits the agent to explore any action that is safe, resulting in more complex representations.

**Our Approach for Explainable-Safe RL.** Rather than presenting the user with a large tree that explains all safety-critical aspects at once, we propose to provide explanations in a *hierarchical* manner. Our method offers a compact, *top-down*, *case-based explanation*. An overview of our method is shown in Fig. 1. First, our approach computes the shield along with its explanations to help users understand how critical the current state is and the risks associated with executing specific actions. Second, the risk of each state is explained using a decision tree constructed at design time. Third, at runtime, additional explanations identify which actions are safe and clarify why others are considered unsafe. Our framework takes as input an abstract world model $\mathcal{M}$ in the form of a Markov Decision Process (MDP), along with a safety specification $\varphi$ in temporal logic.

**At Design-Time.** Based on the model $\mathcal{M}$, we apply value iteration to compute, for every state-action pair $(s, a)$, the risk of violating the safety specification $\varphi$ when executing action $a$ in state $s$. Based on this safety analysis, we categorize the states in $\mathcal{M}$ into four risk categories:

- *Safe*: From safe states, executing any action carries a low risk of violating safety in the future.
- *Critical*: In critical states, some, but not all, actions carry a too-high risk of violating safety.
- *Dangerous*: In dangerous states, all actions carry an unacceptably high risk of violating safety.
- *Unsafe*: States in which safety has already been violated.

**Shield Computation.** From this safety analysis, we first derive the *shield*. The task of the shield is to block actions from the RL agent that pose too high a risk. Thus, in safe states, the shield allows all actions. In critical states, the shield allows all actions whose risk is below the user-defined threshold. If no absolute-safety guarantees are enforced (i.e., the user sets the threshold to a value less than 1, allowing the agent to take some risks), the agent may end up in a dangerous state or even an unsafe one. In dangerous states, all actions induce a risk inevitably above the safety threshold, meaning safety is at risk but, for now, has not been violated. In this case, a fallback shielding strategy must be defined, such as selecting a predefined action (e.g., braking or landing). We follow a common approach and implement a shield that, in such situations, allows only the safest available action [23]. In unsafe states, safety is violated and the shield allows no actions (deadlocks). The resulting shield is a non-deterministic policy mapping states to allowed actions.

**Level 1: Explaining the Risk of States.** The highest-level explanation is represented in the form of a decision tree (DT) that categorizes states into the four risk categories (safe, critical, dangerous, and unsafe), helping users to determine whether the system is currently in a safety-critical situation and to assess the severity of the risk. For a given state, the predicates along the path in the DT provide an explanation for its categorization. *Example: In Fig. 1, a human operator asks for an explanation why the action of* keeping *the velocity is not allowed in the current state s, where the agent is at position* 3 *and an object is at position* 7. *The Level 1 DT explains, that the current state is* critical *because the distance $d$ to the object is $3m \leq d \leq 10m$. Therefore, not all actions are safe to execute.*

**At Runtime.** The user may wish to understand why certain actions are considered safe or unsafe in the current situation. To explain the safety risk of actions, we compute a decision tree that explains the categorization of actions in the current state. If the user requests more detailed information about why a particular action is classified as unsafe, we compute an execution tree that summarizes the potential consequences of executing that action.

**Level 2: Explaining the Risk of Actions.** A Level 2 DT explains the shielding policy for the set of states represented by a leaf in the Level 1 DT. If the current state is categorized as critical in Level 1, the Level 2 DT explains which actions pose a low safety risk and are therefore permitted. If the current state is categorized as dangerous, the Level 2 DT explains what action is the safest one to execute. *Example: Continuing the scenario from Fig. 1, the Level 2 DT explains that, in the current state, only breaking is classified as safe action because the distance to the obstacle is less than 5m. If the distance were greater than 5m, maintaining the current velocity would also be considered safe.*

**Level 3: Explaining the Consequences of Actions.** If executing action $a$ in a given state $s$ is categorized as unsafe, we compute yet another tree that provides evidence for the violation of the property $\varphi$ resulting from executing $a$ in $s$. This final explanation is provided in the form of an *Execution Tree* (ET), which summarizes traces in the MDP $\mathcal{M}$ that start with executing $a$ in $s$, followed by taking only the safest available action thereafter. This tree demonstrates that, even when the safest actions are taken after $a$, the property $\varphi$ is violated with a probability that exceeds the safety threshold. *Example: Returning to the scenario described above, the Level 3 ET demonstrates that, if the speed is maintained in $(p_a = 3, p_o = 7)$, the next state will be $(p_a = 5, p_o = 7)$. From there, even braking would result in an unsafe state $(d = 0)$ within the next two steps.*

To compute the trees, the user can optionally provide predicates to guide the DT learning process. Note that the world model $\mathcal{M}$ and the safety specification $\varphi$ are required in any shielded RL setting. As a result, our method can be easily integrated into any shielded RL application, as it does not require any additional information beyond what is already used to construct the shield.

**Key Contributions of this Paper.** (1) To the best of our knowledge, we introduce the first framework that combines explainability with formal safety guarantees for RL. (2) We provide a method for generating concise, case-based explanations that enable humans to understand the cause of safety violations as they occur. We overcome the non-scalability of a naive approach to explainability-via-DT by the *hierarchical decomposition* of explanations. (3) We evaluate our implementation on several challenging RL benchmarks, showing that the resulting explainable shields are compact and comprehensible, in contrast to traditional shields, which typically involve thousands of states.

## 2 Related Work

Shields were initially introduced in the context of reactive systems [10]. Later, [1] extended the concept to RL, ensuring absolute safety guarantees. In contrast, [23] adopted a quantitative perspective on safety, designing shields that provide probabilistic safety guarantees, thereby permitting the agent to take some calculated risks. These two fundamental concepts of shielded RL assume environments modeled as MDPs with discrete state and action spaces. Many extensions exist [34, 43, 35, 26]. For example, [15] studied shielded RL for *partially observable environments* where only a part of the state can be observed. Shields for *multi-agent systems* have been studied, in both centralized and decentralized settings [18, 29]. Shields for quantitative fairness properties have been studied in [6, 14]. In this work, we make a step toward bridging the gap between formal methods and explainable AI by computing explanations for rigorously computed shields, as introduced by [23].

A comprehensive literature review on explainable RL can be found in [31, 49]. Research directions include human-in-the-loop approaches [40], policy summarization [2, 44], training process visualization [32], and methods that identify performance-critical states [16, 20, 36]. These methods primarily focus on explaining why the agent's decisions are optimal for maximizing the expected reward. The computation of case-based explanations has been explored in [46, 13]. In contrast to explaining the full decision-making intent of the RL policy, our approach explains why certain actions are considered safety-critical.

DTs are widely used in explainable AI due to their intuitive, human-readable structure [17]. In the context of explainable RL, DTs have been constructed to mimic the trained policy of RL agents [9, 30, 47, 19, 38]. In contrast, we compute DTs to explain the shield, resulting in compact explanations that specifically focus on the safety aspects of the problem Furthermore, our approach preserves both optimality and scalability by retaining the RL agent for decision-making. Recently, several works have explored the computation of DTs to represent optimal policies in MDPs [25, 3, 48]. However, these approaches focus on explaining the available actions, not the safety aspects, and are often not scalable to the large environments and non-deterministic policies of the RL context.

## 3 Background

**Markov Decision Processes.** A *Markov decision process (MDP)* [42] is a tuple $\mathcal{M} = \langle \mathcal{S}, s_0, \mathcal{A}, \mathcal{P} \rangle$ where $\mathcal{S}$ is a finite set of states, $s_0 \in \mathcal{S}$ is the initial state, $\mathcal{A}$ is a finite set of actions, and $\mathcal{P} : \mathcal{S} \times \mathcal{A} \to Dist(\mathcal{S})$ is the probabilistic transition function. For all $s \in \mathcal{S}$, the available actions are $\mathcal{A}(s) = \{a \in \mathcal{A} \mid \exists s', \mathcal{P}(s, a)(s') \neq 0\}$ and we assume $|\mathcal{A}(s)| \geq 1$. Let $\mathcal{V} = \{v_1, v_2, \ldots, v_n\}$ be a set of features. A state $s \in \mathcal{S}$ can be represented as a tuple $s = (\overline{v_1}, \ldots, \overline{v_n})$ where $\overline{v_i} = s(v_i) \in \mathbb{Z}$ is the assigned value to feature $v_i$.

Choices in an MDP are resolved via policies. A (memoryless non-randomizing) *non-deterministic policy* is a relation $\pi : \mathcal{S} \to 2^{\mathcal{A}}$. A (memoryless non-randomizing) *deterministic policy* is a function $\pi : \mathcal{S} \to \mathcal{A}$. Applying a deterministic policy $\pi : \mathcal{S} \to \mathcal{A}$ to an MDP $\mathcal{M}$ induces a Markov chain (MC) $\mathcal{M}^{\pi} = \langle \mathcal{S}, s_0, \mathcal{P} \rangle$ with $\mathcal{P} : \mathcal{S} \to Dist(\mathcal{S})$ where all nondeterminism is resolved. A finite *trace* $\tau = \{s_0 s_1 s_2 ... s_n\}$ in a MC is a sequences of states such that $\mathcal{P}(s_i, s_{i+1}) > 0$. The probability mass of such a trace is defined as $\mathbb{P}(s_0 s_1 s_2 ... s_n) = \prod_{0 \leq i < n} \mathcal{P}(s_i, s_{i+1})$.

**Probabilistic Model Checking.** We consider safety properties expressed in Computation Tree Logic (CTL) [8]. Informally, a safety property specifies that 'something bad never happens'. Such properties can represent invariants like 'collisions are never allowed' as well as temporal properties like: 'The agent must not reach an unsafe position within 30 steps.' Probabilistic model checking [8] computes the probability of satisfying a safety property $\varphi$ over a finite or infinite horizon, using adaptations of value iteration, policy iteration, or linear programming. We define the properties below with a bound $n$. For the unbounded horizon, $n = \infty$. For a given MDP $\mathcal{M}$, and a property $\varphi$ in CTL, model checking computes the following probabilities:

- $\mathbb{P}_{\mathcal{M}^{\pi}, \varphi} : \mathcal{S} \times \mathbb{N} \to [0, 1]$ is the expected probability to satisfy $\varphi$ in the MC $\mathcal{M}^{\pi}$ from a state $s \in \mathcal{S}$ within $n$ steps for a deterministic policy $\pi$.
- $\mathbb{P}_{\mathcal{M}, \varphi}^{\min}(s, h) = \min_{\pi} \mathbb{P}_{\mathcal{M}^{\pi}, \varphi}(s, h)$ is the *minimal* expected probability in the MDP $\mathcal{M}$ *over all policies* from a state $s$ within $h$ steps.

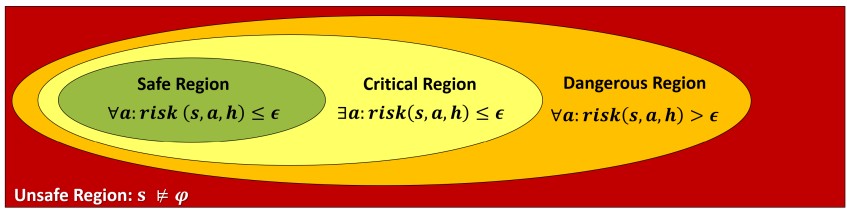

Figure 2: Risk-based Categorization of States.

**Shields.** A *shield* $\pi_{shield} : \mathcal{S} \rightarrow 2^{\mathcal{A}}$ for an MDP $\mathcal{M}$ can be represented as a *non-deterministic policy*. Shields are computed from a safety specification $\varphi$ in CTL and an abstract model of the environment that captures the safety-relevant aspects of the full MDP [23]). Computing the shield using only a safety-relevant fragment of the full MDP enables the scalability of shielded RL. Different types of shields are distinguished by the safety guarantees they provide and the types of used world models. We follow the approach of Jansen et al.[23] and compute shields that offer probabilistic safety guarantees. We discuss the details in Section 4.

**Reinforcement Learning.** In reinforcement learning (RL) [42], an agent learns a task via interactions with an unknown environment modeled by an MDP $\mathcal{M}$ with an associated reward function $\mathcal{R} : \mathcal{S} \rightarrow \mathbb{R}$. In each state $s \in \mathcal{S}$, the agent chooses an action $a \in \mathcal{A}$. The environment then moves to a state $s'$ with probability $\mathcal{P}(s, a, s')$. The return $\text{ret}_\rho$ of an execution $\rho$ is the discounted cumulative reward defined by $\text{ret}_\rho = \Sigma_{t=0}^{\infty} \gamma^t \mathcal{R}(s_t)$, using the discount factor $\gamma \in [0, 1]$. The objective of the agent is to learn an *optimal policy* $\pi^\star : \mathcal{S} \rightarrow \mathcal{A}$ that maximizes the expectation of the return.

**Decision Trees.** We use *decision trees* (DTs) [11] to represent nondeterministic policies in MDPs. A DT over the set of features $\mathcal{V}$ is a tuple $\mathcal{T} = \langle Tr, \Gamma, \rho, \Lambda, \lambda \rangle$ where $Tr$ is a finite, rooted, binary, ordered tree consisting of a set of *inner nodes* $N$ and a set of *leaves* $L$.

The set of *predicates* over $\mathcal{V}$ is denoted by $\Gamma$. We define a set of basic predicates $\Gamma [v_i \sim const]$, where $v_i \in \mathcal{V}$, $const \in \mathbb{Z}$, and $\sim \in \{\leq, <, \geq, >, =\}$. The function $\rho : N \rightarrow \Gamma$ assigns to every inner node $n \in N$ a predicate $p \in \Gamma$. The set $\Lambda$ specifies a set of *labels*. The *labeling function* $\lambda : L \rightarrow \Lambda$ assigns to each leaf $l \in L$, a label $\gamma \in \Lambda$.

A DT $\mathcal{T}$ defines a function $f : \mathbb{Z}^d \rightarrow \mathbb{Z}$, with $d = |\mathcal{V}|$, as follows. Given an input vector (a state in the MDP) $s = (\overline{v_1}, \dots, \overline{v_d}) \in \mathbb{Z}^d$, one follows a unique path $p$ from the root to a leaf $l \in L$ s.t. for each inner node $n \in N$ on the path, the predicate $\rho(n)$ evaluates to true under the substitution $v_i = \overline{v_i}$ iff the first (typically left) child of $n$ lies on $p$. We define $\mathcal{T}(s) = l$ to be the leaf the state $s$ reaches. The value of the function $f$ on $s$ is defined as $f(s) = f(\overline{v_1}, \dots, \overline{v_n}) = \lambda(l)$. We define $l_s : L \rightarrow 2^{\mathcal{S}}$ with $\mathcal{S} = \mathbb{Z}^d$ as the function that maps a leaf $l$ to the set of states whose path ends in $l$.

**Execution Trees.** For a given MC $\mathcal{M} = \langle \mathcal{S}, s_0, \mathcal{P} \rangle$ and a set of finite traces $\Pi = \{\tau_1 \dots \tau_n\}$ of $\mathcal{M}$, an *execution tree* (ET) represents $\tau_1 \dots \tau_n$ of $\mathcal{M}$ in a tree structure. An execution tree $\mathcal{T} = \langle N, E \rangle$ is a rooted tree consisting of a set of nodes $N$ and a set of edges $E : N \rightarrow N$. Nodes are labeled with states $s \in \mathcal{S}$. An edge $e \in E$ corresponds to a transition in $\mathcal{P}$. Each trace $\tau \in \Pi$ defines a path $p$ in $\mathcal{T}$. Thus, each path $p = \{n_1, n_2, \dots n_n\}$ corresponds to a trace $\tau_i = \{s_1, s_2, \dots s_n\} \in \Pi$ such that the node $n_i$ is labeled with the state $s_i$ for $1 \leq i \leq n$.

## 4 Computing Hierarchical Safety Explanations

In this section, we present our algorithm for explainable safe RL in detail. We begin by describing the construction of the shield and the Level 1 DT in Sec. 4.1. In Sec. 4.2, we introduce the Level 2 and Level 3 trees, which provide case-based explanations at runtime. To compute the shield and the explanations, our algorithm takes the safety specification $\varphi$ and an MDP $\mathcal{M} = \langle \mathcal{S}, s_0, \mathcal{A}, \mathcal{P} \rangle$. Optionally, it accepts user-defined predicates to guide the DT learning.

### 4.1 Shield Computation and High-Level Safety Explanations

Executing an action in a given state may carry some risk of violating safety at some point in the future. Based on this risk, states are categorized as *safe*, *critical*, *dangerous*, or *unsafe*.

**Definition 1.** *(Risk of Safety Violation) Given an MDP $\mathcal{M} = \langle \mathcal{S}, s_0, \mathcal{A}, \mathcal{P} \rangle$, a safety property $\varphi$, and a finite horizon $h$, we define the* risk *of violating $\varphi$ from a state $s \in \mathcal{S}$ in the next $h$ steps as a function $risk_{\mathcal{M},\varphi} : \mathcal{S} \times \mathbb{N} \to [0,1]$ as follows:*

$$\forall s \in \mathcal{S} : risk_{\mathcal{M},\varphi}(s,h) = \mathbb{P}^{\min}_{\mathcal{M},\neg\varphi}(s,h).$$

*The* risk *of violating $\varphi$ from a state $s \in \mathcal{S}$ after executing an action $a \in \mathcal{A}$ in the next $h$ steps is defined via the function $risk_{\mathcal{M},\varphi} : \mathcal{S} \times \mathcal{A} \times \mathbb{N} \to [0,1]$ as follows:*

$$\forall s \in \mathcal{S}, \forall a \in \mathcal{A} : risk_{\mathcal{M},\varphi}(s,a,h) = \sum_{s' \in \mathcal{S}} (\mathcal{P}(s,a,s') \cdot risk_{\mathcal{M},\varphi}(s',h-1)).$$

The risk of a state $s$ is defined as the *minimal expected probability* of reaching a state that violates $\varphi$ within the next $h$ steps, quantified over all policies; that is, the probability of a safety violation under the safest available policy. The risk of executing an action $a$ in a given state $s$ is the accumulated, weighted risk of the successor states reached by executing $a$ in $s$. The risk across all states in the state space can be computed using standard probabilistic model checking algorithms, such as value iteration or dynamic programming [8], with tools like PRISM [27], STORM [22], or TEMPEST [34].

**Definition 2.** *(Safety of Actions) Given an MDP $\mathcal{M}$, a safety property $\varphi$, a finite horizon $h$, and a user-defined safety threshold $\epsilon \in [0,1]$. For a given state action pair $(s,a) \in \mathbf{S} \times \mathcal{A}$, action $a$ is called* safe *in $s$ if $risk_{\mathcal{M},\varphi}(s,a,h) \leq \epsilon$. Otherwise, $a$ is* unsafe *in $s$.*

We partition the state space into $\mathcal{S} = \mathcal{S}_s \cup \mathcal{S}_c \cup \mathcal{S}_d \cup \mathcal{S}_u$ according to the risk per state as follows:

**Definition 3.** *(***Safe States** $\mathcal{S}_s$*) $\forall s \in \mathcal{S}$. $s \in \mathcal{S}_s$ iff $\forall a \in \mathcal{A}$: $risk_{\mathcal{M},\varphi}(s,a,h) \leq \epsilon$.*

**Definition 4.** *(***Critical States** $\mathcal{S}_c$*) $\forall s \in \mathcal{S}$. $s \in \mathcal{S}_c$ iff $\exists a \in \mathcal{A}$: $risk_{\mathcal{M},\varphi}(s,a,h) \leq \epsilon$ and $\exists a' \in \mathcal{A}$: $risk_{\mathcal{M},\varphi}(s,a',h) > \epsilon$.*

**Definition 5.** *(***Dangerous States** $\mathcal{S}_d$*) $\forall s \in \mathcal{S}$. $s \in \mathcal{S}_d$ iff $s \models \varphi$ and $\forall a \in \mathcal{A}$: $risk_{\mathcal{M},\varphi}(s,a,h) > \epsilon$.*

**Definition 6.** *(***Unsafe States** $\mathcal{S}_u$*) $\forall s \in \mathcal{S}$. $s \in \mathcal{S}_d$ iff $s \not\models \varphi$.*

The categorization of states is illustrated in Fig. 2. In safe states, all available actions are safe, i.e., they have a risk of at most $\epsilon$. In critical states, only some actions are safe, while others may exceed the safety threshold. In dangerous states, all available actions are unsafe, i.e., each carries a risk of violating the safety property greater than $\epsilon$, although no safety violation has occurred yet. A state is considered unsafe if it violates the safety property.

**Computation of the shield $\pi_{\mathbf{shield}}$.** Using the computed risk function $risk_{\mathcal{M},\varphi}$ and the classification of states, the shielding policy $\pi_{\text{shield}} : \mathcal{S} \to 2^{\mathcal{A}}$ is computed as follows:

$$(s,a) \in \pi_{\text{shield}} \iff \begin{cases} risk_{\mathcal{M},\varphi}(s,a,h) \leq \epsilon, & \text{or} \\ s \in \mathcal{S}_d \text{ and } risk_{\mathcal{M},\varphi}(s,a,h) = \min_{a' \in \mathcal{A}} risk_{\mathcal{M},\varphi}(s,a',h). \end{cases}$$

The shield permits all actions that are classified safe and blocks all unsafe actions. The only exception is in dangerous states, where the shield allows the agent to take the action with the lowest risk. Note that different approaches exist for defining shielding strategies in dangerous states, for example, selecting a predefined fallback action such as halting. Next, we compute the Level-1 DT $\mathcal{T}_{L1}$, which classifies states in the state space according to their risk category, providing an initial explanation of why a given state may be safety-critical.

**Computation of DT $\mathcal{T}_{L1}$.** We construct the decision tree $\mathcal{T}_{L1} = (Tr_{L1}, \Gamma_{L1}, \rho_{L1}, \Lambda_{L1}, \lambda_{L1})$ for a given MDP $\mathcal{M}$, finite horizon $h$, and safety threshold $\epsilon$ through the following steps. First, we define the set of predicates $\Gamma_{L1}$ as follows. $\Gamma_{L1}$ contains the basic predicates $[v_i \sim const]$, where $v_i \in \mathcal{V}, const \in \mathbb{Z}$, and $\sim \in \{\leq, <, \geq, >, =\}$. Additionally, this set can be extended by user-defined predicates to incorporate domain knowledge. For example, predicates measuring the distance of a state $s$ to the unsafe region $S_u$ often enable intuitive explanations of safety-critical dynamics. Such functions could capture, for example, the distance between two agents or the difference between the current temperature and an overheating threshold. These user-defined predicates are of the form $p_i = k_i(v_1, v_2, \ldots, v_n) \sim const_i$ with $k_i : \mathcal{S} \to \mathbb{Z}$. Second, the classification of the states serves as the labels for the leaves of $\mathcal{T}_{L1}$. Thus, $\Lambda_{L1} = \{s, c, d, u\}$. Lastly, we learn $\mathcal{T}_{L1}$ (i.e., its underlying tree $Tr_{L1}$ and functions $\rho_{L1}$ and $\lambda_{L1}$) using standard DT learning algorithms [33, 39]. We learn an exact DT s.t. for all leaves $l$ of $\mathcal{T}_{L1}$, $l_s(l) \subseteq S_x$ for $x \in \{s, c, d, u\}$. Intuitively, every $s \in \mathcal{S}$ is classified exactly, and the tree achieves perfect accuracy.

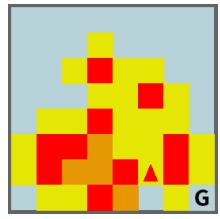
(a) A Frozen Lake environment.

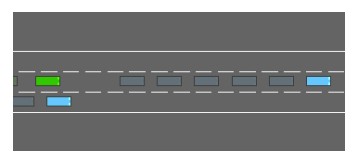
(b) A Highway environment.

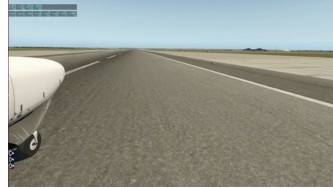
(c) The Taxiing environment.

Figure 3: The RL benchmarks used for evaluation of our approach.

## 4.2 Case-based Safety Explanations

At runtime, our framework provides case-based explanations via two additional trees. Given a current state $s$, a Level 2 decision tree explains which actions the RL agent is allowed to explore from $s$. For a given unsafe action $a$, the Level 3 execution tree explains why executing $a$ is unsafe in $s$.

**Computation of DT $\mathcal{T}_{L2}$.** Let $s$ be the current state. If $s \in \mathcal{S}_s$, all actions are classified as safe. If $s \in \mathcal{S}_u$, safety is already violated. Therefore, we compute action-dependent safety explanations only for critical and dangerous states. In *critical* states $s \in \mathcal{S}_c$, some actions are safe and some are unsafe (i.e., $\exists a\colon risk_{\mathcal{M},\varphi}(s,a,h) \le \epsilon$ and $\exists a'\colon risk_{\mathcal{M},\varphi}(s,a',h) > \epsilon$.). A $\mathcal{T}_{L2}$ for a critical state $s$ explains why actions are safe or unsafe in $s$. In *dangerous states*, all actions are unsafe, i.e., $\forall a\colon risk_{\mathcal{M},\varphi}(s,a,h) > \epsilon$. In such states, our shield allows the agent to explore the safest available action (which still exceeds the safety threshold $\epsilon$). Then, the $\mathcal{T}_{L2}$ provides an explanation for why the selected action is considered the safest among the available options.

Let $s$ be the current state and $l = \mathcal{T}_{L1}(s)$ be the leaf node reached by state $s$ in $\mathcal{T}_{L1}$. If $s \in \mathcal{S}_c$ or $s \in \mathcal{S}_d$, we compute a Level-2 DT $\mathcal{T}_{L2}^l = (Tr_{L2}^l, \Gamma_{L2}^l, \rho_{L2}^l, \Lambda_{L2}^l, \lambda_{L2}^l)$ over the states $l_s(l)$ as follows. The set of predicates $\Gamma_{L2}^l$ is equal to $\Gamma_{L1}$. The set of labels $\Lambda_{L2}^l$ is defined as sets over actions, since the $\mathcal{T}_{L2}^l$ explains which actions are allowed by the shield, i.e., $\Lambda_{L2}^l = 2^{\mathcal{A}}$.

**Computation of ET $\mathcal{T}_{L3}$.** Given a current state $s_{cur}$ and an unsafe action $a_u$, the ET $\mathcal{T}_{L3}$ explains that executing $a_u$ in $s_{cur}$ leads to unsafe states with a probability greater than $\epsilon$. The tree provides this explanation by representing a set of traces that start in a state reached after executing $a_u$ in $s_{cur}$ and ending in an unsafe state, and that provide enough evidence to demonstrate that $a_u$ is unsafe. For a given state pair $(s_{cur}, a_u)$, we compute the $\mathcal{T}_{L3}^{s_{cur},a_u}$ as follows. First, the MDP $\mathcal{M}$ is transformed into an MC $\mathcal{M}^{\pi} = \langle \mathcal{S}, s_0, \mathcal{P} \rangle$ via the following policy $\pi$:

$$\forall s : \pi(s) = \begin{cases} a_u & \text{if } s = s_{cur}, \text{and} \\ \arg\min_{a \in \mathcal{A}} risk_{\mathcal{M},\varphi}(s,a,h) & \text{otherwise.} \end{cases}$$

Thus, the policy $\pi$ picks in any state the safest available action. Only in the current state $s_{cur}$, the policy $\pi$ selects the action $a_u$. Therefore, traces sampled from this policy demonstrate possible consequences of executing $a_u$ in $s_{cur}$, if afterward only the safest actions were executed. We call a finite trace $\tau = \{s_0, \ldots, s_n\}$ *unsafe* if $s_n \in \mathcal{S}_u$. We compute a set of traces $\Pi = \{\tau_1, \tau_2, \ldots \tau_k\}$ that start in state $s_{cur}$ in the MC $\mathcal{M}^{\pi}$ such that the probability mass of these traces exceeds the safety threshold $\epsilon$, i.e. $\sum_{\tau \in \Pi} \mathbb{P}(\tau) \ge \epsilon$. We follow the algorithm from [21] to compute the most probable traces in the MC $\mathcal{M}^{\pi}$ that are unsafe and their probability mass exceeds the threshold $\epsilon$. The algorithm uses the recursive enumeration algorithm for computing the $k$ shortest paths on a weighted digraph [24], substituting probabilities in $\mathcal{M}^{\pi}$ with distances. This algorithm computes the shortest paths recursively until the probability mass of the traces is greater than $\epsilon$. The resulting traces are prefix-merged to construct the execution tree $\mathcal{T}_{L3}^{(s_{cur},a_u)}$.

## 5 Experimental Evaluation

In this section, we present the experimental evaluation of our approach. We consider the size of a shield $|\pi_{shield}|$ as the size of the lookup table and the size of a tree $|\mathcal{T}|$ as the number of its nodes. The tree size serves as our metric for evaluating explainability. The model checking queries were computed using TEMPEST [34], and the DT representations of shields using DTCONTROL [5]. All experiments were conducted on a laptop with an Intel® Core™ i7-11800H CPU at 2.3 GHz with 32

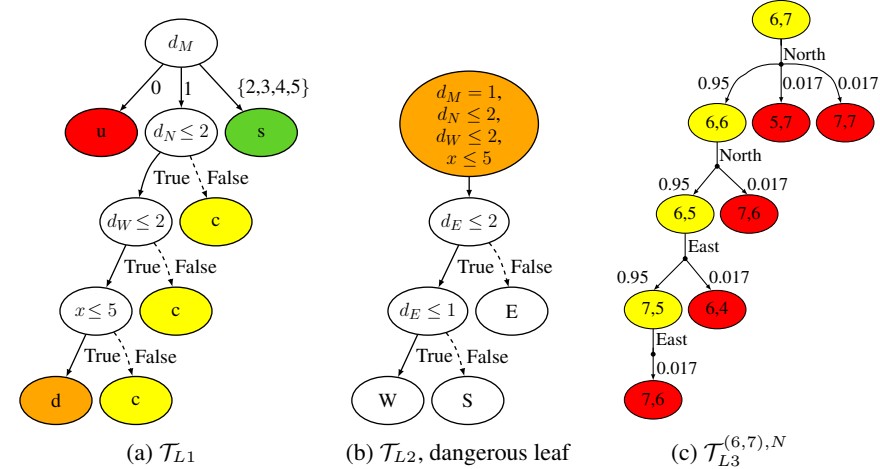

(a) $\mathcal{T}_{L1}$       (b) $\mathcal{T}_{L2}$, dangerous leaf       (c) $\mathcal{T}_{L3}^{(6,7),N}$

Figure 4: Exemplary explanations for the Frozen Lake environment.

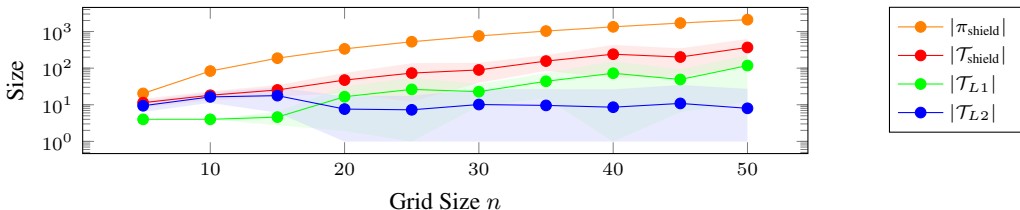

Figure 5: Shield and tree sizes for different configurations of the Frozen Lake environment.

GB of RAM. All details of the experimental setup can be found in the Appendix. We provide the implementation as supplementary material.

## 5.1 Frozen Lake

We performed a first set of experiments using the Farama Frozen Lake environment [45]. A Frozen Lake environment is an $n \times n$ grid, consisting of blue slippery tiles and holes. The agent can move in cardinal directions, where every movement carries a probability of $0.05$ of slipping into a different direction. The RL agent has to reach the goal **G** while not falling into a hole. The shield preventing the agent from falling into a hole is computed with a horizon $h = \infty$ and a risk threshold $\epsilon = 0.075$. The state classification of an example environment is shown in Fig. 3a. Red tiles indicate unsafe states (holes), orange tiles indicate dangerous states, yellow tiles indicate critical states, and blue tiles indicate safe states. The set of predicates $\Gamma = \{x, y, d_M, d_N, d_E, d_S, d_W\}$ consists of the agent's coordinates $(x, y)$, the Manhattan distance $d_M$ to the nearest hole, and the distance $d_a$ to the nearest hole in the cardinal direction $a \in \{N, S, E, W\}$.

**Explaining Safety.** Fig. 4a shows $\mathcal{T}_{L1}$ for the environment depicted in Fig. 3a. $\mathcal{T}_{L1}$ can concisely capture the states in which the shield needs to interfere, by using the user-defined distance predicates. Fig. 4b shows $\mathcal{T}_{L2}$ for the dangerous states in $\mathcal{T}_{L1}$. The tree summarizes that, to exit a dangerous state, the agent should move east if the distance to a hole in that direction permits it; otherwise, it needs to move to the nearest safe state. Finally, Fig. 4c shows why moving to the *North* at position $(6, 7)$ is not allowed. The represented traces in $\mathcal{T}_{L3}^{(6,7),N}$ have a probability mass of $0.08$ of reaching an unsafe state, which exceeds the allowed risk of $0.075$.

**Results.** We evaluated the scalability of our approach using randomly generated instances of Frozen Lake environments of increasing size $n \times n$ with $n \in \{5, 10, \ldots, 50\}$. Per size, we generate 10 random instances and compare the sizes of the computed shields and tree representations. Fig. 5 shows the average sizes of the shielding policy $|\pi_{shield}|$, the shield represented as one single tree $|\mathcal{T}_{shield}|$, and the trees $|\mathcal{T}_{L1}|$, and $|\mathcal{T}_{L2}|$ over grid size $n \times n$. The results show that, using our approach, we obtain DTs that are several orders of magnitude smaller than the shielding policy. The average

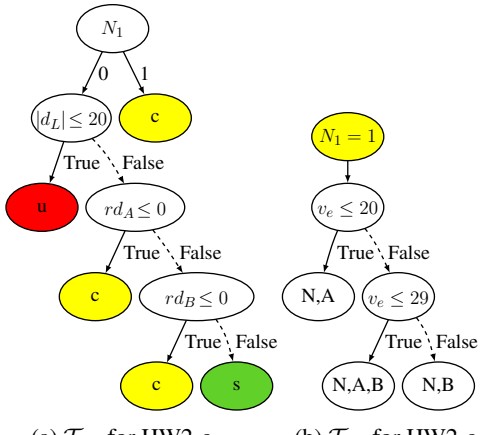

| | $|\pi_{\text{shield}}|$ | $|\mathcal{T}_{\text{shield}}|$ | $|\mathcal{T}_{L1}|$ | $|\mathcal{T}_{L2}|$ |
|---|---|---|---|---|
| HW2-f | 358 | 7 | 7 | 3 |
| HW2-c | 3553 | 113 | 9 | 33 |
| HW3-f | 8355 | 18 | 17 | 2.5 |
| HW3-c | 232216 | 555 | 57 | 45.1 |
| TaxiNet | 1105 | 39 | 21 | 6 |
| TaxiNet | 1107 | 37 | 33 | 5 |

(a) $\mathcal{T}_{L1}$ for HW2-c  (b) $\mathcal{T}_{L2}$ for HW2-c    (c) Sizes of the shielding policy and the DTs

Figure 6: Exemplary DTs for HW2-c and the comparison of the sizes of the shield and decision trees.

time for computing the shield ranges from $0.15$s for $n = 5$ to $15.4s$ for $n = 50$. The average time to learn the DTs ($\mathcal{T}_{L1}$ or $\mathcal{T}_{L2}$) is approximately 2s for all grid sizes.

## 5.2 Highway Cruise Control

We conducted our second set of experiments in the Farama Highway environment [28], illustrated in Fig. 3b. In this environment, the agent controls a self-driving car operating on a highway populated with other vehicles. The agent's actions are: switching lanes ($S_L$, $S_R$), accelerating ($A$), braking ($B$), or performing no operation ($N$). The agent's goal is to reach the end of the road without collisions. Additionally, it receives a positive reward for driving in the leftmost lane. We compute a shield that ensures collision avoidance by enforcing a safe distance of 20m. In this example, the shield prohibits the agent from taking any risks (i.e., $\epsilon = 0$) with $h = \infty$. We consider two parameters: two or three lanes (HW2/HW3) and whether the agent can change its velocity (-f(ixed)/-c(hangeable)). This results in the scenarios HW2-f, HW2-c, HW3-f, and HW3-c. We discuss the results of HW2-c. Appendix A.2 provides further results and precise predicate definitions. The set of predicates $\Gamma = \{d_1, l_e, d_L, N_1, rd_A, rd_B\}$ includes: the distance $d_1$ to the nearest vehicle in lane 1; the current lane of the ego vehicle $l_e$; the distance $d_L$ to the closest vehicle in the ego vehicle's lane; $N_1$ indicating whether the ego vehicle is next to a vehicle in lane 1; and final predicates encoding the remaining distance to other cars when the agent accelerates ($rd_A$) or brakes ($rd_B$).

**Explaining Safety.** Fig. 6a shows the $\mathcal{T}_{L1}$, and 6b shows the $\mathcal{T}_{L2}$ for the topmost critical node. The trees explain that if the ego car is next to a car in the adjacent lane, switching lanes is not allowed. Moreover, the agent may only accelerate or brake within its velocity bounds of $20 - 29$m/s.

**Results.** Table 6c shows the size of the shielding policy and the average sizes of DTs of different levels and scenarios. The results show, as expected, that representing shields via trees instead of lookup tables yields a more compact representation. Furthermore, using the hierarchical explanations $\mathcal{T}_{L1}$ and $\mathcal{T}_{L2}$ is, in most cases, more compact than using a single tree. The time for computing the shield ranges from $0.1$s for HW2-f to $50s$ for HW3-c. The time to learn the DTs ranges from $1.3$s for $\mathcal{T}_{L1}$ of HW2-f to $17.4$s of $\mathcal{T}_{L1}$ for HW3-f.

## 5.3 Boeing Taxinet

For our final set of experiments, we applied our approach to an autonomous taxiing environment [41], as shown in Fig. 3c. In this environment, the agent must steer an aircraft to ensure it remains aligned with the centerline. Depending on the current heading, the aircraft's position relative to the centerline changes. The safety objective is to avoid exceeding a heading of $20°$ and a maximum deviation of 0.8m from the centerline. The shield is computed with a horizon $h = \infty$ and a risk threshold $\epsilon = 0.0$. The set of predicates $\Gamma = \{he, cte, |he|, |cte|, d\}$ consists of the heading error $he$, centerline error $cte$, their absolute values, and the distance $d$ to the maximum allowed centerline error, depending on the current heading. The set of actions $\mathcal{A} = L, N, R$ allows the agent to steer left ($L$) or right ($R$) in steps of $5°$, or to take no action ($N$). We have applied our approach

to two different instances and show the resulting sizes for the shields and the DTs in Table 6c. The results again show that our approach yields compact representations of the shielding policy. We provide the corresponding DTs and further discussion in the Appendix.

## 6  Conclusion & Future Work

We presented a method for "explainably safe" RL via proposed explainable shields. Our approach provides case-based explanations of the shielding policy and a hierarchy of decision trees that explains the risks associated with states and actions, as well as the consequences of executing unsafe actions. Our experiments show the capability of our method in providing small trees even for complex scenarios. In future work, we will explore automated methods for predicate generation to reduce reliance on user input. Besides, we plan to investigate the potential of generative AI to render case-based explanations understandable to non-specialists. Furthermore, a user study on the understandability of our explanations is also of interest. Lastly, we plan to extend our approach by computing compact explanations for states that the trained RL policy visits frequently. As a trained RL policy rarely operates over the entire state space, focusing explanations on the most frequently visited states will provide clearer insights into the safety-relevant aspects of the agent's behavior.

## Acknowledgments

Bettina Könighofer and Stefan Pranger were supported by the State Government of Styria, Austria - Department Zukunftsfonds Steiermark and by the Austrian Science Fund (FWF) 10.55776/COE12. This research has furthermore received funding from the European Union under Grant Agreement No. 101171844, ERC project Intelligence-Oriented Verification&Controller Synthesis (InOVationCS), and from the European Union's Horizon Europe program under Grant Agreement No. 101212818, Robustifying Generative AI Through Human-Centric Intergration of Neural and Symbolic Methods (RobustifAI). Views and opinions expressed are, however, those of the authors only and do not necessarily reflect those of the European Union or European Research Executive Agency. Neither the European Union nor the granting authority can be held responsible for them. This research has also received funding from the MUNI Award in Science and Humanities MUNI/I/1757/2021 of the Grant Agency of Masaryk University.

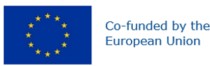 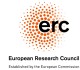 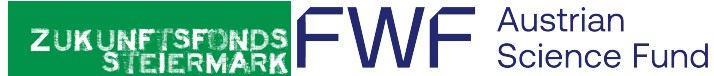

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

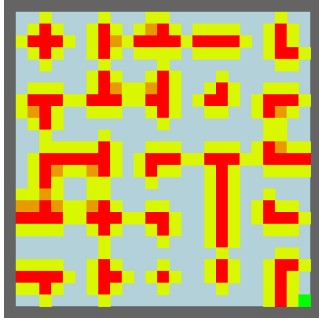

Figure 7: A randomly generated Frozen Lake environment with approximately 15% of holes.

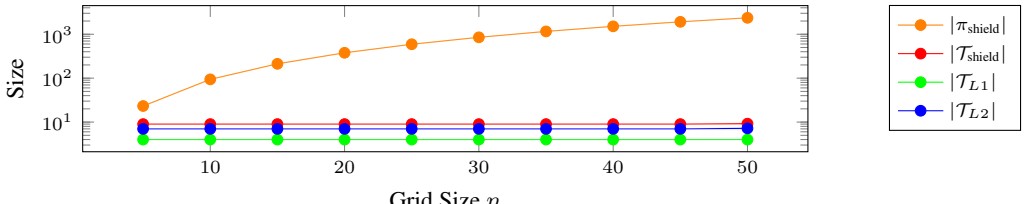

Figure 8: Size comparison between the different approaches using random Frozen Lake instances with approximately 4% of holes.

# A    Appendix

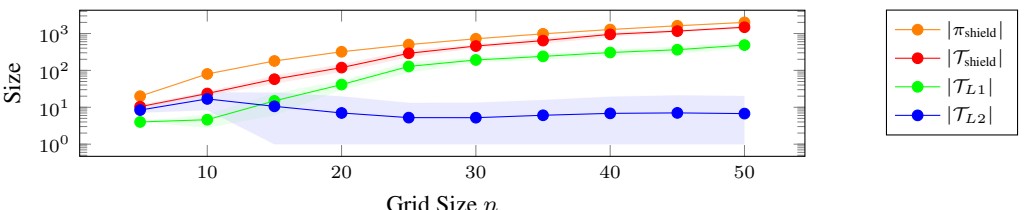

Figure 9: Size comparison between the different approaches using random Frozen Lake instances with approximately 20% of holes.

## A.1    Additional Results —Frozen Lake

**Scalability.** For the experiments on the scalability of our approach, we randomly generated instances of the Frozen Lake environment of various sizes. To ensure that all environments have a meaningful structure, we divide the overall size $N \times N$ of the instance into tiles of size $5 \times 5$. In the center of each tile, we place a randomly shaped connected hole of size 0 to 5. Fig 7 shows such a randomly generated environment. Red areas represent holes, orange areas the dangerous states, and yellow the critical states. For the experiment presented in the main part of the paper, Figure 5, we create the holes in such a way that the expected value of holes is $15\%$ of all fields. The figure shows that the size of a DT is already smaller than the size of the original shield. Our method reduces the size further, even for large problem instances.

In Figure 8, we perform the same experiment with randomly generated instances, but only allow holes of sizes 0 to 2, which leads to an expected value of $4\%$ of all fields being holes. In this case, all DTs are small and remain small for very large instances.

On the other hand, when creating instances with $20\%$ of the entire grid being holes, the size of the DTs grows. This is caused by the fact that holes are close to each other because of their size. When a field is surrounded by several holes, it becomes dangerous instead of critical. Therefore, having many holes interacting with each other causes the overall problem to become more complicated and makes it harder to distinguish critical from dangerous states.

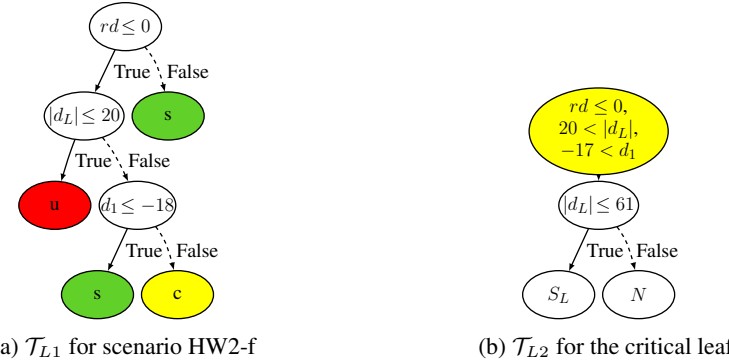

(a) $\mathcal{T}_{L1}$ for scenario HW2-f        (b) $\mathcal{T}_{L2}$ for the critical leaf

Figure 10: Decision Trees for scenario HW2-f

## A.2   Additional Results —Highway Cruise Control

In this section, we discuss the remaining scenarios for the Highway environment and elaborate on the suggested predicates.

**HW2-f.** In the base scenario, the highway has two lanes, and all vehicles are driving at a fixed speed. The agent is slightly faster with a value of $v_d$ than the vehicle on the right lane. The available actions are *switch left ($S_L$), switch right ($S_R$),* and *noop (N)*. All actions succeed with probability 1. The feature set of the environment is $\mathcal{V} = \{d_1, l_e\}$, the distance to the vehicle on lane 1, which is the rightmost lane, and the current lane of the ego vehicle. We extend the set of basic predicates $\Gamma = \{d_1, l_e, d_L, N_1, rd\}$ with the following predicates:

- $d_L := (d_1$ if $l_e = 1$ else $100)$ is the distance to the vehicle on the current lane of the ego vehicle. If there is no vehicle on the lane, the distance is set to a high value.
- $rd := d_1 - v_d - 20$ is the remaining distance until action has to be taken immediately. It can be seen as the distance to the safety buffer around the other vehicle. To avoid comparison with the constant $v_d$, we subtract it.

The DT $\mathcal{T}_{L1}$ describing this instance is depicted in Fig. 10a. It consists of 7 nodes. The DT investigates whether the remaining distance to the car on lane 1 remains large assuming no action is taken. If this is the case, every action is allowed. Otherwise, the DT compares to the safety threshold to filter out unsafe states. Lastly, when the agent is in front of the vehicle on lane 1, it may perform any action. Otherwise, as the root tests that the agent is close to the vehicle on lane 1, the state is critical. Figure 10b shows $\mathcal{T}_{L2}$ for the critical state. If the agent is on the left lane, shown by a high lane distance, it may not switch to the right lane. Otherwise, it has to switch to the left lane to avoid a collision. Computation of $\mathcal{T}_{L1}$ was completed within less than 1.4 seconds, $\mathcal{T}_{L2}$ computation required on average 1.6 seconds and shield creation 0.1 seconds.

**HW2-c.** In this scenario, the agent is allowed to change its speed and accelerate or decelerate by a fixed value of $\Delta_v$. Therefore, the set of features is extended with the velocity of the ego vehicle and the vehicle on lane 1 to $\mathcal{V} = \{d_1, l_e, v_e, v_1\}$. The agent has the additional actions *accelerate (A)* and *brake (B)*. We provide the predicates $\Gamma = \{d_1, l_e, v_e, v_1, d_L, N_1, rd_A, rd_B\}$ with the addition of:

- $rd_A := (d_1 - (v_e + \Delta_v - v_1) - 20$ if $l_e = 1$ else $100)$ is the remaining distance until the safety requirement is violated, assuming the agent accelerates and the other vehicle is on the same lane.
- $rd_B := (d_1 - (v_e - \Delta_v - v_1) + 20$ if $l_e = 1$ else $100)$ is the remaining distance until the safety requirement is violated, assuming the agent decelerates and the agent is on lane 1.
- $N_1 := (d_1 \in [-20 + v_d, 20 + vd])$ which captures whether in the next time step, after the ego car has approached the other vehicle by $v_d$ space units, it will be within the safety zone of the other vehicle. In this case, the ego vehicle is considered to be next to the other vehicle.

The new predicates are a refinement of $rd$ to allow for a better understanding of the behavior of different actions.

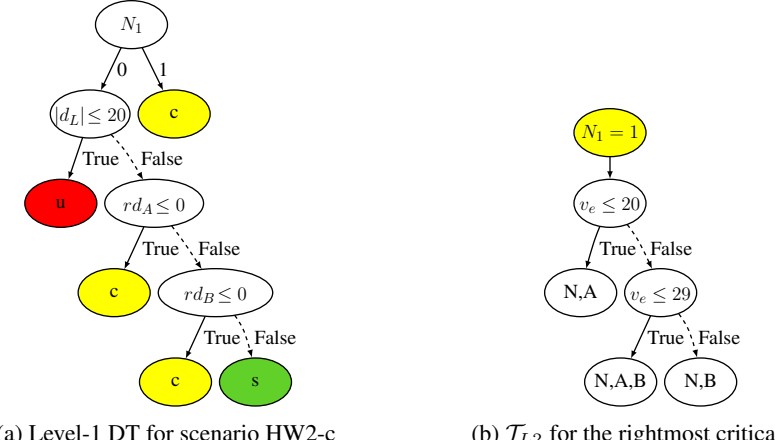

(a) Level-1 DT for scenario HW2-c

(b) $\mathcal{T}_{L2}$ for the rightmost critical leaf

Figure 11: DTs for scenario HW2-c

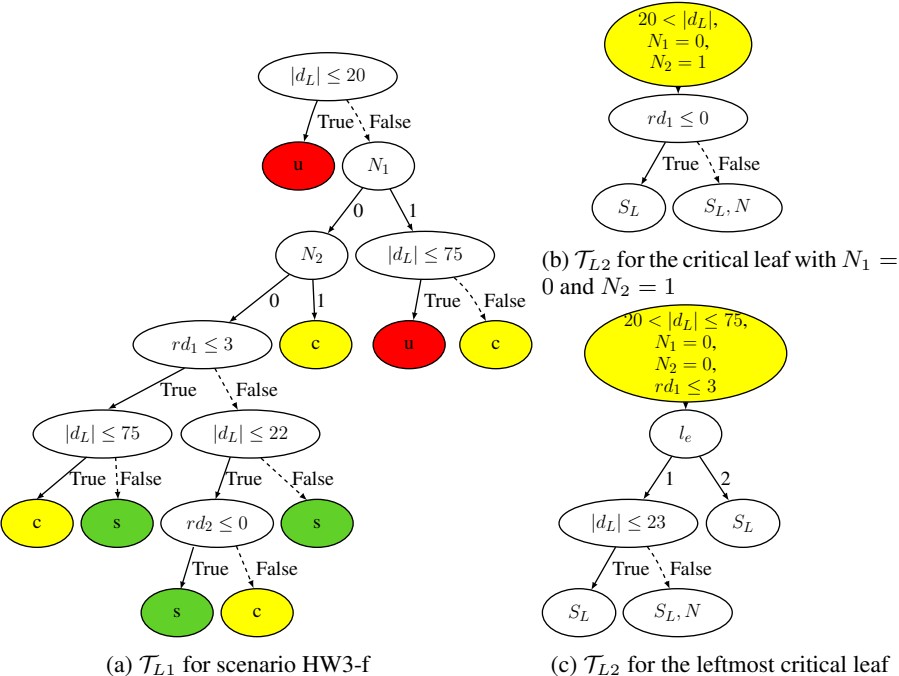

(a) $\mathcal{T}_{L1}$ for scenario HW3-f

(b) $\mathcal{T}_{L2}$ for the critical leaf with $N_1 = 0$ and $N_2 = 1$

(c) $\mathcal{T}_{L2}$ for the leftmost critical leaf

Figure 12: DTs for scenario HW3-f

$\mathcal{T}_{L1}$ is depicted in Figure 11a. It explains that when the ego vehicle is next to another vehicle, some actions are prohibited. Furthermore, when the current distance falls below the safety distance, the state is unsafe. In all other cases, the criticality depends on whether acceleration and braking are safely possible. In this example, no dangerous states exist as the environment is deterministic. Level-2 DTs consist, on average, of 33 nodes. This is caused by one tree having 81 nodes while all others remain small, similar to the one in Figure 6b, which is also depicted in Figure 11b. Computing $\mathcal{T}_{L1}$ required 1.3 seconds, the average time for $\mathcal{T}_{L2}$ was again 1.3 seconds, and creation of the shield took 0.7 seconds.

**HW3-f.** This scenario is an extension of HW2-f with an additional lane. All vehicles drive at a fixed speed, with the ego vehicle being slightly faster by a value of $v_d$. The set of features is extended with the distance to the vehicle on lane 2 $\mathcal{V} = \{d_1, d_2, l_e\}$. We extend the set of predicates $\Gamma = \{d_1, d_2, l_e, d_L, N_1, N_2, rd, co\}$ We define $N_1$ and $N_2$ accordingly for the vehicles on lanes 1 and 2, and define $rd_1$ and $rd_2$ as $rd$ for the vehicles on lanes 1 and 2, respectively.

The Level-1 DT is depicted in Figure 12a. It shows that the classification of the states depends on the distance to the vehicle in front and whether the agent is next to another vehicle. If the vehicle is next to one on lane 1 and has a vehicle close in front, it cannot avoid a collision.

On average, the Level-2 DTs have a size of 2.5 nodes, with the largest one having 5 nodes. Examples are shown in Figures 12b and 12c. Figure 12b corresponds to the critical leaf where $N_1 = 0$ and $N_2 = 1$. $N_2 = 1$ already encapsulates that the agent is currently on the middle lane. The available actions then depend on whether it is too close to the vehicle on lane 1, which disallows action $N$. In any case, switching to the leftmost lane is allowed. For the leftmost critical leaf, which is shown in Figure 12c, we know that the agent is not next to any vehicle. However, the current remaining distance to the vehicle on lane 1 is low. Therefore, if the agent is on lane 2 it should switch left to avoid a scenario where overtaking the vehicle on lane 2 is no longer possible. If it is on lane 1, the available actions depend to the distance of the vehicle on lane 1. Computation of $\mathcal{T}_{L1}$ was completed within 1.8 seconds, $\mathcal{T}_{L2}$ required, on average, less than 1.3 seconds, and shield creation less than 2.7 seconds.

**HW3-c.** In the last scenario, we consider a highway with three lanes and allow the ego vehicle to change its velocity. The state space $\mathcal{V}$ is extended with *pa*, which shows what action was taken in the previous step. It is required to precisely model the Highway RL environment, where the speed after no operation or switching lanes also depends on the action of the previous time step.

We allow all predicates of the previous cases and add a new predicate *sbd*, safe braking distance, which compares the braking distance ($bd$) needed for the ego vehicle to brake until it is at most as fast as the vehicle in front of it, to the remaining distance: $sbd := (d_L \geq bd)$. It is an extension of $rd_B$ as $rd_B$ considers switching lanes as a possible action, whereas *rbd* only allows braking. We obtain a Level-1 DT of slightly below 70 nodes, which is depicted in Figure 13. The size of the tree results from the special cases in which the ego vehicle can switch to the right-most lane and break before hitting the vehicle in front of it, or switch back to the middle lane. Introducing a predicate for these special cases reduces the tree size drastically but requires more detailed domain knowledge. This shows the need for the automatic finding of intelligent predicates. Similar behavior can be observed for the Level-2 DTs, which have an average size of 36 nodes. Computation of the $\mathcal{T}_{L1}$ required 17.4 seconds. Level-2 DTs are computed on average in less than 3.6 seconds. Shield computation was performed within 50 seconds.

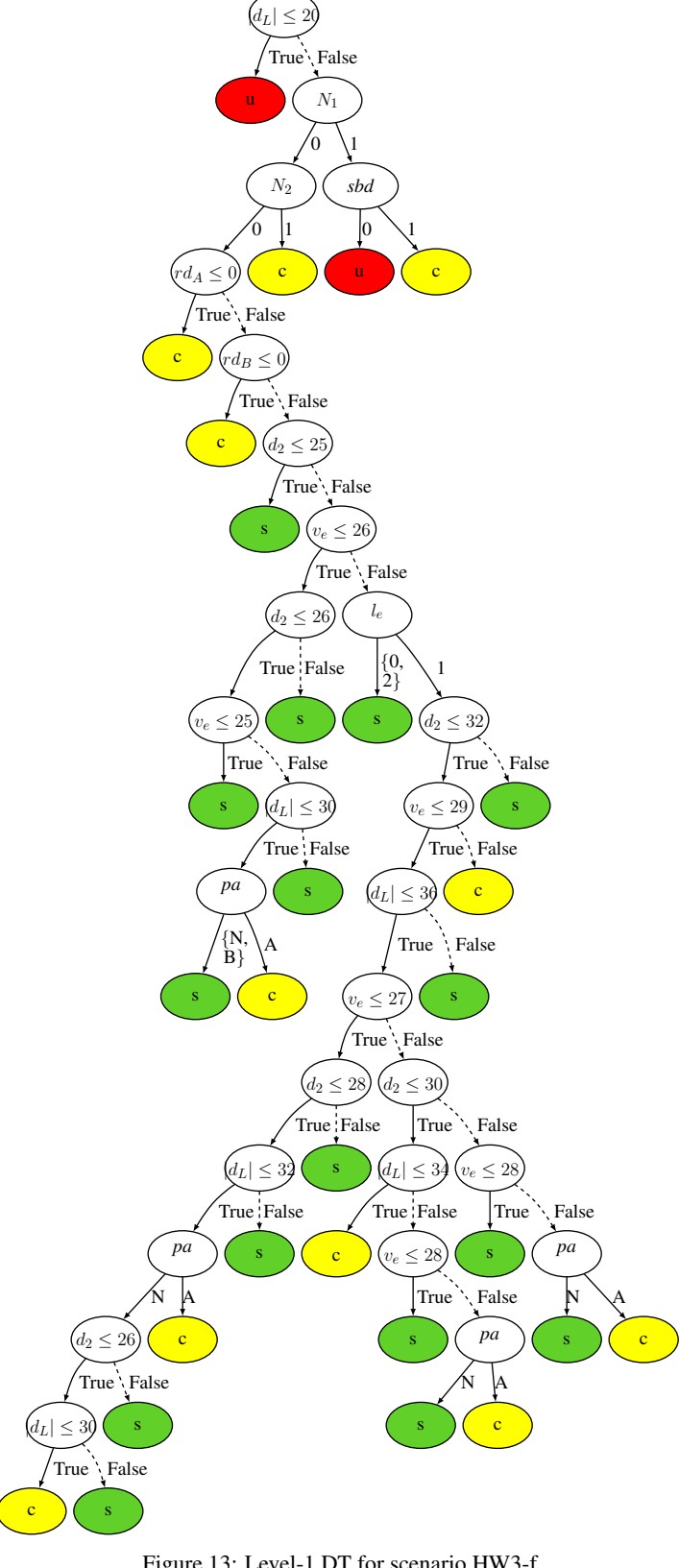

Figure 13: Level-1 DT for scenario HW3-f

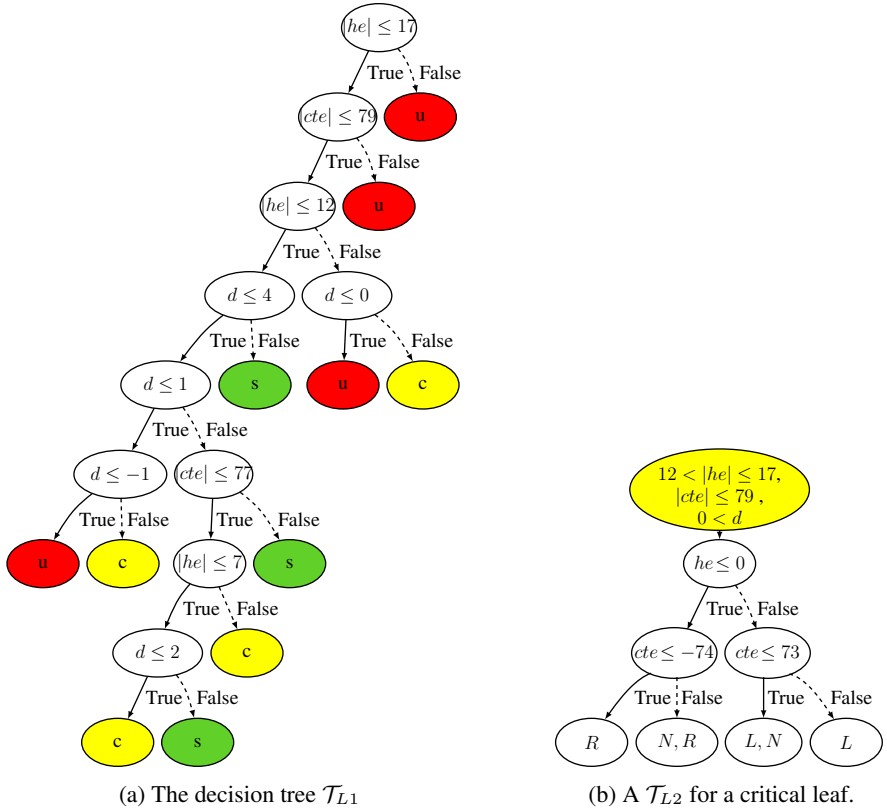

(a) The decision tree $\mathcal{T}_{L1}$

(b) A $\mathcal{T}_{L2}$ for a critical leaf.

Figure 14: Decision trees for the taxiing environment.

### A.3 Additional Results —Boeing Taxinet

In this section, we discuss the results for two different autonomous taxiing environments: a model where steering always follows a deterministic update and a slippery model, where steering might cause the airplane to slip, causing its heading to change by $10°$ instead of $5°$. Initially, the airplane is positioned at the centerline with a heading of $0°$. Depending on its current heading, the airplane moves $\frac{he}{5}$ decimeter per time step.

**Taxiing.** In this base scenario the agent will only be restricted by the shield if either it would reach the maximum heading error of $20°$ or in case it would diverge from the centerline by $80$ decimeter. The decision tree $\mathcal{T}_{L1}$ and a $\mathcal{T}_{L2}$ for a critical leaf are shown in Fig. 14a and 14b, respectively. The level-1 DT $\mathcal{T}_{L1}$ nicely shows how user-defined predicates can help explain the shield in a concise way. After classifying the states in which the property has already been violated, the $\mathcal{T}_{L1}$ differentiates between states in which the heading error is smaller or larger than $12°$. It can then immediately make use of the defined distance function. In case the heading error is still small and the distance is sufficiently large, all states are safe and no interference is needed. If the distance is $\leq 40$ decimeter $\mathcal{T}_{L1}$ classifies the different scenarios depending on concrete distance, heading and centerline errors. If the heading error exceeds $12°$ and safety has not yet been violated, the shield has to interfere. The restrictions the shield imposes are shown in 14b. This $\mathcal{T}_{L2}$ nicely shows how a compact representation allows us to legibly explain how the shield interferes.

The synthesis time for the shielding policy is 0.19 seconds and the time needed to compute either a $\mathcal{T}_{L1}$ or a $\mathcal{T}_{L2}$ tree is 1.5 seconds.

**Taxiing on Slippery Ground.** In this second experiment, the aircraft might slip an additional $5°$ with a probability of $10\%$ in case it is steered to the left or to the right. The resulting decision trees are shown in 15. The Level-1 DT $\mathcal{T}_{L1}$, shown in 15a, captures this by more finely distinguishing between the current centerline error and distance to the maximum centerline error. Similarly to the Level-1 DT from the previous experiment, it classifies all states in which the heading error is

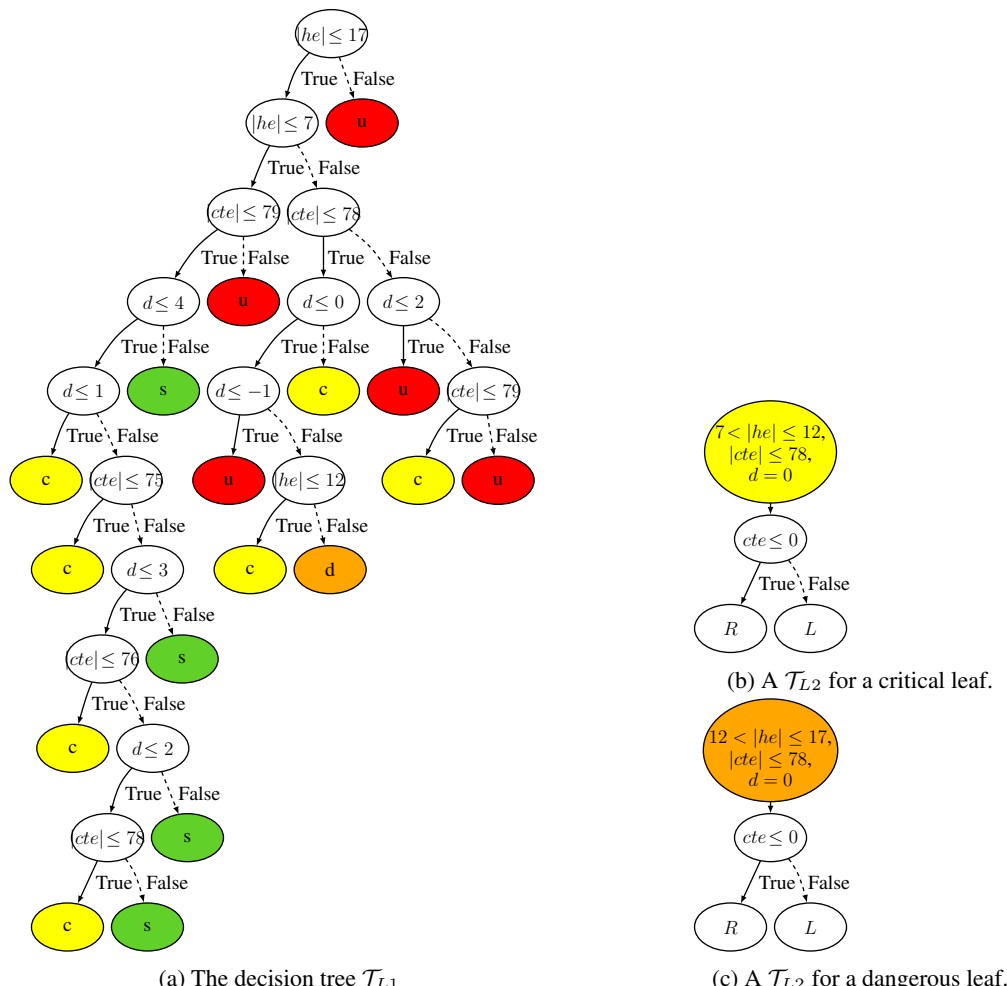

(a) The decision tree $\mathcal{T}_{L1}$

(b) A $\mathcal{T}_{L2}$ for a critical leaf.

(c) A $\mathcal{T}_{L2}$ for a dangerous leaf.

Figure 15: Decision trees for the slippery taxiing environment.

sufficiently small, $\leq 7°$, and the distance to the maximum centerline error is large enough, as safe. In case the heading error is small, but the centerline error is close to the maximum allowed value, $\mathcal{T}_{L1}$ distinguishes between safe and critical states by capturing the different values for the distance and centerline error. In case the heading error is $> 7°$, the decision tree $\mathcal{T}_{L1}$ has to classify states that are either critical, dangerous, or are already violating the safety specification. Fig. 15b and Fig. 15c show two examples for $\mathcal{T}_{L2}$. We want to highlight these two examples for $\mathcal{T}_{L2}$, as they show how our approach is able to explain both how the shield has to interfere in order to enforce the safety specification, as well as when the safety specification cannot be adhered to anymore and only safest actions can be allowed by the shield anymore.

As above, the synthesis time for the shielding policy is 0.19 seconds and the time needed to compute either a $\mathcal{T}_{L1}$ or a $\mathcal{T}_{L2}$ tree is 1.5 seconds.

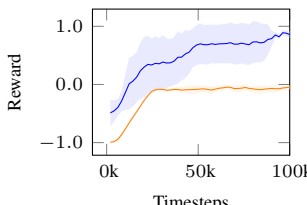 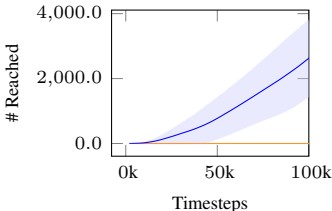 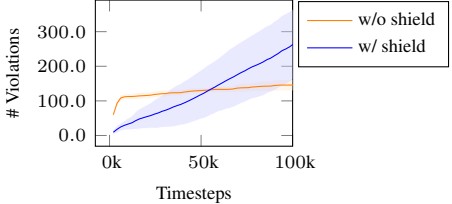

(a) Rewards for Frozen Lake environment

(b) Accumulated number of successful episodes

(c) Accumulated safety violations

# B  Supplementary Material

We have trained agents on the Frozen Lake environment from Fig. 3a and the four different settings in the Highway environment using the implementations from Stable-Baselines3 [37]. The library provides a *MaskablePPO* implementation, which is perfectly suited for safe RL via shielding. We have compared shielded training with training without a shield using *PPO*. All training runs have been conducted using the default parameters.

Due to the simulator for the Boeing TaxiNet environment being closed-source, we were not able to train agents for these problem instances.

## B.1  Frozen Lake Environment

The task of the agent in this environment is to reach the goal state without falling into a hole. The agent can move in any cardinal direction and will succeed with a probability of $0.95$. With a probability of $0.05$ it will slip in any of the other cardinal directions where it is not obstructed by a wall. Upon reaching the goal the agent receives a reward of $1$, otherwise, when the agent falls into a hole it receives a negative reward of $-1$.

Fig. 16a, 16b, and 16c show the training results averaged over 5 runs. These results show that the shield enables the agent to finish its task, while training without the shield does not succeed. When training without a shield, the agent is not able to explore the critical areas around the holes and therefore stays at the safe area near the initial position only. We want to remark that due to the stochasticity exhibited in this environment, complete safety cannot be guaranteed.

## B.2  Highway Environment

The task of the agent in this environment is to reach the end of the highway as fast as possible without causing a crash. The agent can switch lanes, do nothing, or change its velocity, if it is not fixed. The agent receives a reward of $1.0$ for driving on the rightmost lane, a reward of $0.5$ on the leftmost lane, and a reward of $0.75$ for driving on the middle lane in the environments HW3-f and HW3-c. If the agent crashes into another car, it receives a negative reward of $-1.0$.

We show the training results, averaged over 5 runs in Figures 17a to 17h. We want to use the example of environment HW3-c to highlight the need for accurate world models: Due to the world model not being 100% accurate, we have exhibited a small number of crashes in the shielded training run.

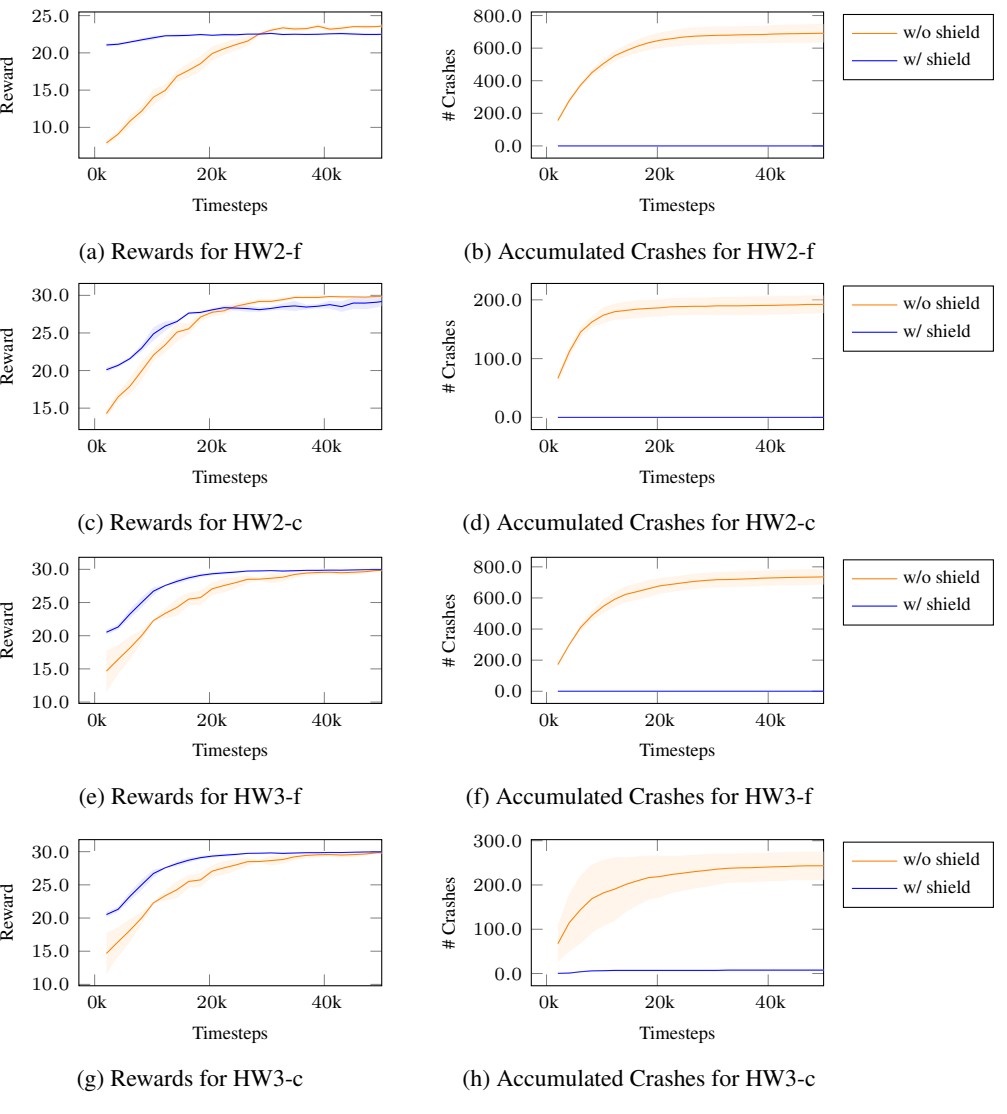

(a) Rewards for HW2-f

(b) Accumulated Crashes for HW2-f

(c) Rewards for HW2-c

(d) Accumulated Crashes for HW2-c

(e) Rewards for HW3-f

(f) Accumulated Crashes for HW3-f

(g) Rewards for HW3-c

(h) Accumulated Crashes for HW3-c

Figure 17

