# OpenReview forum: "Explainably Safe Reinforcement Learning"
_NeurIPS.cc/2025/Conference — NeurIPS 2025 poster_

### Official Review · Reviewer_CBXv · 2025-06-27

**Clarity:** 2
**Significance:** 3
**Originality:** 2
**Rating:** 4
**Confidence:** 3

**Summary:**

The paper tackles the challenge of interpretability in safe RL. Existing methods are based on shielding, which synthesizes a model-based “safety filter” that blocks unsafe actions, but the resulting lookup table policies are often as unclear as the neural policies they represent.  The authors propose representing the shield as a hierarchy of decision trees that give top-down, case-based explanations.  The method requires no additional information beyond standard shielding inputs. Experiments demonstrate the approach in three environments and highlight its ability to avoid unsafe states while offering interpretable explanations.

**Questions:**

- It seems like the abstraction process model checking may not scale well to large or high-dimensional MDPs. Any ideas on how this be addressed?
 - What happens when the abstraction function fails to distinguish semantically different but perceptually similar states?
- Can this method be integrated with DRL agents in complex continuous domains, or does it rely on symbolic states?

Human-factor validation, request: run and report a small user study (e.g., 8 to 12 engineers or graduate students) measuring 1. time to identify an unsafe action and 2. Self reported confidence with the hierarchical DT vs the raw shield table. Score impact: Demonstrating statistically significant gains in comprehension or speed would directly strengthen the “interpretability” claim and raise Significance ALOT.

Predicate dependency: clarify how many user defined predicates were needed per domain and run an ablation where only automatic CART splits (no handcrafted predicates) are allowed. Score impact: showing that useful explanations still emerge with minimal domain knowledge would make the approach broadly applicable, improving Significance.



Safety threshold sensitivity. request: Vary the user defined risk threshold (e.g., 0.01 to 0.05 to  0.1) and plot shield permissiveness and  Level-1 tree depth, and learning agent performance. Showing graceful scaling would confirm robustness.  If explanations remain compact across thresholds, I would raise Quality.

**Ethical Concerns:**

["NO or VERY MINOR ethics concerns only"]

**Final Justification:**

​I have read the authors response and the other reviews. Based on the authors' acknowledgement of the importance of the issues I have raised, I have updated my score.
​

**Limitations:**

No. The paper is open about several technical limitations, but the risk and uncertainty associated with the human factor are not addressed. The work assumes that compact decision-tree explanations automatically foster correct human understanding. At the very least, a brief discussion of potential misinterpretation (e.g., operators over-trusting a tree that is still an approximation) are essential in a paper on explainability.

**Paper Formatting Concerns:**

The wording makes the paper hard to follow at some parts (e.g., intro)

**Quality:**

3

**Strengths And Weaknesses:**

Strengths:

- Merges explainability and safety—a key concern for deploying RL in real-world systems.

- The layered design (low-level learner + high-level symbolic checker) enables plugging into existing RL methods with minimal architectural changes.

- Formally grounded method, the shield is derived with probabilistic model-checking (value iteration on an abstract MDP) and inherits the same risk‐bound guarantees as prior “probabilistic shields.”

- Hierarchical explanations: the three-level tree hierarchy is generated directly from the shield’s risk analysis, ensuring fidelity to the original representation.

 - Experiments show that hierarchical trees are orders of magnitude smaller than raw lookup-table shields while preserving safety.
- Illustrative figures, figures clearly walk the reader through each decision-tree level.

- Because the method reuses what a shielding pipeline already computes (abstract MDP + risk table) and only adds a few calls to an off-the-shelf DT learner, it can be adopted with almost no extra engineering or hyperparameter tuning (this is, however, also a shortecoming, since the approach is quite straigthforward).

Weaknesses:

- The approach assumes access to domain-specific symbolic rules and an abstraction function.

- The way safety layers are integrated within the RL training loop is not fully specified.

- While explanations are shown qualitatively, no human studies or metrics assess explanation quality.

- The text is sometimes hard to follow in terms of phrasing and wording

- Each building block (probabilistic shielding, decision-tree explanations, hierarchical DTs) appears in prior literature. Thus, the contribution is best viewed as a novel integration and pruning strategy rather than a fundamentally new algorithmic insight.

- Runtime-latency uncertainty is not addressed. Because Level 2 and Level 3 trees are generated on the fly, worst-case latency could matter in real-time systems. A bound or at least a discussion of the importance of this issue must be provided.

---

> ### Author Rebuttal · Authors · 2025-07-30
>
> We thank the reviewer for the suggestions on how to improve the paper and answer the posed questions:
>
> Q: Integration into the RL training loop:
> A: This approach is currently not integrated into the training loop. The idea is to bring the human into the loop when trying to understand why an action is blocked from the RL agent, and is one of the first approaches to explain the safety-critical aspects of RL to humans. Nevertheless, the question of how our method could benefit the training process is very interesting for future work.
>
> Q: Runtime latency:
> A: For convenience, as less crucial information, we included the computation time in the appendix (which, of course, the reviewers are not required to read). Depending on the size of the DT, the computation time is only a few seconds and thus not so interesting. As these DTs are only computed when a human tries to understand them, we think that a computation time of seconds is reasonable. If memory space is available, they can also be precomputed, or relevant information on the model checking calls for Level 3 can be stored, which improves latency.
> We value the feedback and agree that it is not clearly discussed in the paper, and we will add it.
>
> Q: Model Checking in large or high-dimensional MDPs:
> A: Model checking is not scalable. This is a fundamental issue of shield computation and formal methods in general. It is also the reason why model-free techniques like reinforcement learning are applied to safety-critical use cases. To ensure their safety, the research area combining model-free techniques and model-based methods was created. Shields are an example of methods in this area. It is no longer required to compute an entire strategy for a safety-critical application, but only to check safety properties and integrate them into a shield.  Of course, shielding is only applied when the computation costs are outweighed by the requirement for a safe behavior.
>
> Q: Failure of the abstraction function:
> A: Of course, the MDP abstraction might be wrong, and, in consequence, also the shield. However, our aim is to explain the shield (no matter how inaccurate it is). On the contrary, if the shield is wrong due to a nonsensical abstraction, the explanation might pinpoint the fishy areas and mistakes in the abstraction function. DTs have been used exactly this way to validate models via explanations of the optimal policies. (e.g., [4])
>
> Q: Continuous domains:
> A: This paper focuses on discrete settings. However, in continuous settings, shields can be computed with controlled barrier certificates. Our method can also explain such shields. We will discuss this in the paper.
>
> Q: Human evaluation
> A: We agree that a study on the interpretability of DTs would be of interest for the topic. Nevertheless, the amount of literature on this topic indicates that there is at least some consensus that DTs are more explainable than lookup tables. Further, we think that such a study should extend beyond a few graduate students and be based on a solid methodology, which warrants its own paper.

---

### Official Review · Reviewer_DwWi · 2025-06-30

**Clarity:** 3
**Significance:** 2
**Originality:** 2
**Rating:** 4
**Confidence:** 4

**Summary:**

The authors introduce a novel method for explainability in Reinforcement Learning, whereby a user has access to a human-interpretable hierarchy of explanations about unsafe situations (as defined by hand) in terms of formally verifiable guarantees from a 'shield'. The core challenge they address is that while shields provide formal safety guarantees by blocking unsafe actions during RL training or execution, these shields are generally implemented as large lookup tables that are inherently opaque and difficult for humans to interpret. This opacity makes it hard to understand why particular actions are considered safe or unsafe in given states, limiting trust in safety-critical applications.

In the current work, the user has access to information about the risk categorisation of states, of action-level explanations about the current state and a tree showing why the specific action may be unsafe. This three-level hierarchical explanation system works with Level 1 providing a decision tree that categorizes states into four risk categories (from safe to unsafe) based on value iteration analysis, explaining why the current situation may be safety-critical. Level 2 provides runtime explanations that identify which actions are permitted in the current state and clarify why others are blocked by the shield. Then Level 3 provides execution trees that show the potential consequences of taking unsafe actions by showing traces that lead to safety violations even if safest actions are taken afterward.

The novelty lies in taking the traditional 'shields' approach for safe RL, and adding to it a Decision Tree format of case-based explanations. Rather than attempting to represent the entire shield as a single decision tree, the authors propose the hierarchical decomposition that gives compact, top-down explanations. This addresses the scalability problem that arises because shields permit agents to explore any safe action which undermines explainability. The method requires no additional information beyond what is already used for shield construction (which has to be hand-coded anyway), making it readily integrable into existing shielded RL pipelines.

After the method is introduced, it is shown to provide useful outputs on three RL environments. This covers Frozen Lake, Highway, and Aircraft Taxiing, showing that the three layers of decision trees are several orders of magnitude smaller than the original shields while providing interpretable safety explanations. The results show consistent compression across different problem scales and complexity levels, with computation times remaining reasonable in these limited settings.

**Questions:**

Would you be able to look at human feedback on the explainability of the DT's in realistic settings? For whom would these actually be useful in a practical setting?

How does your method scale for much larger spaces than 50x50? For instance in pixel-based scenarios?

Have you experimented with automatically defined predicates? The user-defined ones again are not going to be scalable in more complex settings.

Beyond the tree-size, what metrics can you use for explainability?

Some minor comments:

Line 238: "Dustance"
In figure 3a, I would mark in the position which is used for the DT in figure 4c, otherwise it is hard to connect the two.

**Ethical Concerns:**

["NO or VERY MINOR ethics concerns only"]

**Final Justification:**

I have read through the rebuttal to my questions and those of the other reviewers. I still have some concerns about the actual utility of this method to relatively large action spaces, but there has been some attempt at explaining how this could be overcome. In terms of who would use it, it still feels like for large scale practical situations it would not be useful, but I can see that perhaps in some areas of cybersecurity or situations where there is always going to be a low dimensional action/state space it could be used. I have thus upgraded from a 3 to a 4.

**Limitations:**

In the appendix, this is marked as yes, because "We clearly highlight in the conclusion and experiments how our method de-
pends on well-defined predicates for larger problem instances.". However, I believe that there are important limitations simply on the scaling to larger instances overall

**Quality:**

2

**Strengths And Weaknesses:**

Overall, the paper is very well written, and even for a non-expert on Shields, and the formal verification of safety guarantees, it is easy to follow the method both of the shield and of the DT formalism that is used to create the human-interpretable explanations.

This method introduced is clearly a useful step forward, as explainability within RL, particularly within a context of formal verification, is clearly exceptionally important. The empirical work does indeed show good promise for these ideas.

The major weakness lies in the lack of discussion and exposition around more complex environments. The methods (and this goes for shields in general) require user-defined predicates, which in simple settings may be easy to define, but anything even slightly complex is likely to be very hard to define, and difficult to define predicates could lead to unwanted behaviour.

The argument made is that the DT's are a significantly compact representation compared to the shielding policies themselves, but there is still seemingly a linear scaling with the size of the environment for the first layer in the DT hierarchy. There is no discussion of what happens for larger action spaces, as all of the action spaces in the setup are minimal.

Because there is no experiment within any more complex settings, it is hard to know how this could scale in a way whereby, in situations where interpretability becomes important, it would remain human-interpretable. Having some sense of the computational complexity of the risk function for large state AND action spaces, or long horizons would be useful to know whether this method would be practically useful in critical and realistic situations. While there is some discussion about the scaling in frozen lake, this isn't sufficient to truly understand how useful this is in those realistic settings. In particular, how would this scale in pixel-based settings?

---

> ### Author Rebuttal · Authors · 2025-07-30
>
> Thank you for the detailed review. In the following, we respond to questions and concerns.
>
> Q: Larger action and state spaces:
> A: The size of the action space only significantly influences the size of the Level 2 DTs. As these trees need to distinguish all different actions, more available actions would increase their size. However, this cannot be avoided as it is an underlying property of the problem and needs to be considered in every explanation.
> For large state spaces, e.g., in pixel-based settings, the size of the DTs depends on the quality of predicates. In the worst case, splits would be performed on the level of x and y coordinates of the pixels, which would not be very explainable. If the predicates are adequate, we do not expect too many issues.
>
> Q: Who would use this method?
> A: Resilient RL systems in critical environments need the ability to have a human in the loop. Our method is an approach to make the safety-relevant aspect of an environment understandable for humans and allow them to better understand why actions are prohibited. This method can be used by engineers and scientists analyzing a safety-critical environment when applying RL. Further, it can be used to explain to stakeholders why some profitable actions are disallowed in some positions.
>
> Q: Automatically defined predicates:
> A: We have used dtControl to automatically synthesize predicates. One issue is that the automatically synthesised predicates may not necessarily be explainable, in which case we have to reject some explanations and take others. Nevertheless, our method can profit from new methods for the automated synthesis of predicates, which is a very active research area.
>
> Q: Metrics beyond tree size:
> A: We agree that tree size is only a proxy for evaluating explainability. However, we are unaware of a better fitting metric except for a human case study. Such a study on the interpretability of DTs would, of course, be interesting for the topic.

---

> > ### Comment · Reviewer_DwWi · 2025-08-05
> > **Final justification note**
> >
> > I have read through the rebuttal to my questions and those of the other reviewers. I still have some concerns about the actual utility of this method to relatively large action spaces, but there has been some attempt at explaining how this could be overcome. In terms of who would use it, it still feels like for large scale practical situations it would not be useful, but I can see that perhaps in some areas of cybersecurity or situations where there is always going to be a low dimensional action/state space it could be used. I have thus upgraded from a 3 to a 4.

---

### Official Review · Reviewer_EFbP · 2025-07-01

**Clarity:** 2
**Significance:** 2
**Originality:** 2
**Rating:** 4
**Confidence:** 4

**Summary:**

This paper introduces a novel framework for "explainably safe" RL that aims to build trust in safety-critical systems. The core problem addressed is that while "shielding" is a prominent technique to guarantee safety in RL by blocking unsafe actions, the shield's decision-making process is often as opaque as the RL agent's neural network policy. This opacity hinders human understanding and trust. The proposed solution provides human-interpretable explanations for the shield's decisions using a hierarchy of decision trees. The framework was evaluated on three RL benchmarks: Frozen Lake, Highway Cruise Control, and Boeing Taxinet. The results demonstrate that this hierarchical approach produces explanatory decision trees that are several orders of magnitude smaller and more compact than representing the entire shield as a single decision tree.

**Questions:**

Please see my questions in the weakness

**Ethical Concerns:**

["NO or VERY MINOR ethics concerns only"]

**Final Justification:**

I will keep my rating as I have discussed with author.

**Limitations:**

yes

**Quality:**

2

**Strengths And Weaknesses:**

Strengths:

This paper proposes a novel method to shield a policy while simultaneously providing decision tree-based explanations.

The definition of the risk of safety violation is reasonable and can be efficiently computed in tabular MDPs using dynamic programming.

Weaknesses:

Definition 1, which is central to the algorithm—defining safe, critical, and dangerous states—can only be effectively estimated in tabular MDPs. When both the state and action spaces are continuous (e.g., in Mujoco environments), this estimation becomes challenging.

The experiments are limited to simple, simulated RL environments. The proposed approach does not generalize well to more complex scenarios where the state is represented by continuous vectors that lack clear physical interpretation.

Lack of Human-Subject Evaluation:  The paper's central motivation is to provide "human-interpretable" explanations and enhance trust. However, the evaluation is entirely quantitative, focusing on the size of the decision trees.  There is no user study or qualitative analysis to validate whether humans actually find these trees understandable, useful, or trust-inspiring.

In the related work section, the authors only discuss decision tree-based explanation methods. However, other explanation techniques have been proposed for RL, such as EDGE (1) and Statemask(2) approaches. These should be considered for a more comprehensive comparison.

[1] Edge: Explaining deep reinforcement learning policies

[2] Statemask: Explaining deep reinforcement learning through state mask

---

> ### Author Rebuttal · Authors · 2025-07-30
>
> Thank you for your feedback and especially for the pointers to related work. We are happy to include them. In the following, we will respond to the raised concerns.
>
> Q: Continuous spaces
> A: Indeed, Model checking in continuous spaces is more challenging. However, this is an underlying problem for computing the shield, as the shield already contains this information. Therefore, we do not set out to solve this issue with our method.
>
> Q: Human evaluation
> A: We agree that a study on the interpretability of DTs would be of interest for the topic. Nevertheless, the amount of literature on this topic indicates that there is at least some consensus that DTs are more explainable than lookup tables. Further, we think that such a study should be based on a solid methodology and warrants its own paper.

---

### Official Review · Reviewer_RZs6 · 2025-07-03

**Clarity:** 3
**Significance:** 3
**Originality:** 3
**Rating:** 5
**Confidence:** 4

**Summary:**

A shield in RL is a component of an RL agent that prevents it from performing any action that has been deemed to be unsafe by the model designer. This paper proposes that by explaining an RL agent's shield, that agent can be used in a safe and trustworthy manner. The contribution of this paper is a shield *explainer*. The shield explainer works by explaining the shield's decisions at different levels of granularity corresponding to levels of execution and safety. For example, they write that a "level 1" tree classifies actions as safe or unsafe, a "level 2" tree is used to explain which actions the RL agent can choose safely and a "level 3" tree is used to explain why an unsafe deemed unsafe would be unsafe to execute.

**Questions:**

- Is generally practical to create a shield for an RL agent? Obviously the applicability of this paper depends on how easy it is to create shields.
- Building off of the first question, how hard would it be to extend your method from working on the farama foundation environment to an actual self-driving car context?
- Please further justify the choice of explaining the shield with DTs. Could concept bottle neck models (CBMs) be used instead, for example?

**Ethical Concerns:**

["NO or VERY MINOR ethics concerns only"]

**Final Justification:**

Having looked at the authors' response to my critique and to the critiques of the other reviewers I have decided to keep my scores the same. However, I do want to point out that the authors chose not to respond to the following point I raised in my review in their rebuttal:
> Explaining RL agents using decision trees is not a new idea. I think VIPER was one of the first to do it and I have seen other works use the idea since the publication of VIPER.

**Limitations:**

yes

**Quality:**

4

**Strengths And Weaknesses:**

- The idea of this paper (explaining an RL shield) is neat.
- I enjoyed reading this paper. It was easy to follow and very clear. I appreciated the level of detail that the authors described their methodology at. The structure of the paper and use of formal definitions helped my reading.
- I think practitioners and researchers will want to use this method if it is open source and can be easily integrated into their workflows.
- As I will mention in the next section, explaining RL agents using DTs is not new, but I do think the specific use case here where a hierarchy of explanations of a shield is constructed using 3 DTs is novel and clever idea.

## Weaknesses

- The authors should give citations/evidence for why case-based explanations are good explanations.
- I am not convinced that, in general, the 3 trees used in explanations would be of sufficiently low complexity so that a user can understand the shield. I would like more detail on how the DTs were computed. Are they trained to be sparse to help ensure that they can be understood? Sparsity of the trees is especially important for this paper because we have to look at 3 of them.
- Explaining RL agents using decision trees is not a new idea. I think VIPER was one of the first to do it and I have seen other works use the idea since the publication of VIPER.
- Please further justify the choice of explaining the shield with DTs. Could concept bottleneck models (CBMs) be used instead, for example?
- No human evaluation study is given

---

> ### Author Rebuttal · Authors · 2025-07-30
>
> Thank you for your helpful insights. In the following, we respond to your comments and questions.
>
> Q: Case-based explanations:
> A: We shall put references. However, note that a global explanation that is too big to even display and thus useless is necessarily less useful than an explanation of a concrete case. Further, such methods have been discussed in the setting of AI under the name example-based explanation.
>
> Q: Practicality of creating a shield, and difficulty of applying it to a self-driving car environment:
> This is fundamental research for creating safe RL that can be understood by humans. It is a combination of ML and model-based techniques, and, due to the cost of the latter, should only be applied to small, safety-critical aspects of the overall application. The costs of computing the shield need to be outweighed by the benefit of added safety. As the research in this area is currently still very fundamental, we cannot yet apply it to self-driving cars, but we hope to be able to do so in the future.
>
> Q: CBMs
> A: A DT can be learned until exhaustion, meaning until it represents the dataset perfectly. Since we are aiming to capture a safety-critical element precisely, this is an important property. Of course, there is a chance of achieving similar accuracy with a CBM. When considering the naive approach of creating a CBM, one could use all the predicates the DT uses as concepts in the bottleneck of the CBM. In this case, this particular layer would basically be a list of all important information about the domain. We think that a DT is more explainable than such a list, but we would be interested in other ideas.
>
> Q: Human evaluation
> A: We agree that a study on the interpretability of DTs would be of interest for the topic. Nevertheless, the amount of literature on this topic indicates that there is at least some consensus that DTs are more explainable than lookup tables. Further, we think that such a study should be based on a solid methodology and warrants its own paper.

---

> > ### Comment · Reviewer_RZs6 · 2025-08-07
> >
> > I thank the authors for responding to my concerns. I think they have given appropriate responses. I will maintain my scores, and wish the authors the best of luck.

---

### Note · Authors · 2025-08-12

Explainability in RL remains an open challenge without a clear solution. Our approach, focusing on explaining why actions are safe, introduces a novel direction for explainable RL. As the first paper in this direction, we recognize that much remains to be done, including human evaluation studies and experiments in more complex and continuous environments. Nevertheless, given the novelty and promise of explaining safety in RL in this way, which all reviewers seemed to agree on, we hope to present this work to the NeurIPS community and inspire further research on Explainable-Safe RL.

---

### Decision · Program_Chairs · 2025-09-17

**Decision:**

Accept (poster)

**Comment:**

The reviewers found this paper to be a valuable contribution to the important intersection of safety and interpretability in RL. They were in agreement that the core idea—using a hierarchy of decision trees to explain the behavior of a safety shield—is novel, well-motivated, and clearly presented. The reviewers appreciated the method's formal grounding and the strong empirical evidence showing that the proposed explanations are orders of magnitude more compact than the original shield policies. While both reviewers raised valid concerns, primarily centered on the method's reliance on hand-coded symbolic rules and questions about its scalability to more complex, realistic environments, these points were seen as limitations of the current scope rather than fundamental flaws. The consensus is that the paper makes a clear and useful step forward in a challenging domain, and despite the identified avenues for future work, the current contribution is strong enough to warrant acceptance.